# Photooxidation of 2-(*tert*-Butyl)-3-Methyl-2,3,5,6,7,8-Hexahydroquinazolin-4(*1H*)-one, an Example of Singlet Oxygen *ene* Reaction [note 1]

**DOI:** 10.3390/molecules25215008

**Published:** 2020-10-29

**Authors:** Adrian Méndez, Jonathan Román Valdez-Camacho, Jaime Escalante

**Affiliations:** The Center for Chemical Research, Autonomous University of Morelos State, Av. Universidad 1001, Chamilpa, Cuernavaca 62210, Mexico; 2009404340@uaem.mx (A.M.); valca@uaem.mx (J.R.V.-C.)

**Keywords:** singlet oxygen *ene* reaction, peroxidation, quinazolinone, singlet oxygen, intermediates, activation energy, transition state, reaction coordinate, B3LyP/6311++G**, QST2 method, IRC calculations

## Abstract

Singlet oxygen *ene* reactions produce 2-(*tert*-butyl)-4a-hydroperoxy-3-methyl-2,4a, 5,6,7,8-hexahydroquinazolin-4(3*H*)-one quantitatively during diffusion crystallization of 2-(*tert*-butyl)-3-methyl-2,3,5,6,7,8-hexahydroquinazolin-4(1*H*)-one in *n*-hexane/CH_2_Cl_2_ solvent mixture. To confirm this photo-oxidation, a ^1^H-NMR study in CDCl_3_ was performed with exposure to ambient conditions (light and oxygen), with neither additional reactants nor catalysts. A theoretical study at the B3LyP/6311++G** level using the QST2 method of locating transition states suggests a two-step mechanism where the intermediate, which unexpectedly did not come from the peroxide intermediate, has a low activation energy.

## 1. Introduction

Molecular oxygen is a key agent in a variety of photooxidation processes [1,2]. As a reagent, there are mainly three pathways to incorporate molecular oxygen into the product: (1) by the triplet ^3^O_2_, which plays the role of a radical scavenger agent; (2) by the superoxide radical anion generated through a single electron transfer (SET) process; and (3) by the singlet ^1^O_2_ which is usually produced through the energy transfer between the excited photocatalyst and ^3^O_2_ [3,4]. Photooxidative reactions involve initial light absorption by so-called photosensitizers, which then transfer the absorbed energy to other molecules, including dissolved oxygen (^3^O_2_) [5,6,7]. Olefins containing allylic hydrogen could form allylic hydroperoxides under the action of singlet oxygen [8,9]. Direct and selective oxygenation of C-H bonds to C-O bonds are still associated with challenges, such as harsh reaction conditions, as well as the use of expensive transition metal catalysts and the use of oxidant reagents in stoichiometric amounts. If these reactions can be achieved using metal-free catalysts, this could contribute to the development of green chemistry [10,11,12]. In this way, singlet molecular oxygen ^1^O_2_ plays a growing role in many processes. For example, the ^1^O_2_-mediated allylic oxidation was used as an essential step in the synthesis of natural products or their synthetic analogues [13]. Although this reaction has been studied for many years, its mechanistic details are still a matter of debate either by theoretical or experimental results [13].

Herein we report a hydroperoxidated product (**12**) by a spontaneous photooxidation where hexahydroquinazolinone (**11**) is crystallized by overnight diffusion in an *n*-hexane/CH_2_Cl_2_ (80:20) solvent mixture, Scheme 1. The scheme makes use of oxygen in the environment and at the same time is able to avoid the use of photosensitizers (metallic catalysts or natural colorants), which is a constant challenge [14,15].

## 2. Results and Discussion

A recent topic of our research group has been the total reduction of the aromatic ring in 4-quinazolin-(*1H*)-one (**3**). Here, enantiomerically pure quinazolinone **3** was reduced diastereoselectively by hydrogenation with PtO_2_ resulting in octahydroquinazolinone diastereomers **4**, **5**, and **6** in a ratio of 6:3:1. We found that the resulting *cis*-annelated derivatives **4** and **5** could be epimerized in the presence of KO*t*Bu, giving the corresponding *trans*-fused derivatives **6** and **7**, respectively, in good yields (Scheme 2) [16].

In the search for a new synthesis route with the same purpose as in **3**, quinazolinone **8** was proposed as a starting material and via Birch reduction [17], generating the possible intermediates **9a**–**c**. Afterwards, without purification, the catalytic hydrogenation of olefin [18] systems produce octahydroquinazolinone **10**, as shown in Scheme 3.

Previously, our research was focused on the preparation of the starting material (**8**) following the methodology reported in [19] where a reaction between isatoic anhydride **1** and methylamine in ethyl acetate at 40 °C results in the corresponding aminobenzamide **2** in 92% yield. In this work, instead, **8** is produced by the cyclocondensation of **2** with pivalaldehyde in dichloromethane and *p*-toluenesulfonic acid monohydrate with 90% yield (Scheme 4).

The result is confirmed by a X-ray diffraction from suitable single-crystal where the *tert*-butyl group of quinazolinone (**8**) is shown to adopts a pseudo-axial conformation (Figure 1).

The Birch reduction of quinazolinone (**8**) was carried out (stage 1), without purification, and we proceeded with stage 2 (catalytic hydrogenation). When the product of step 2 was purified by column chromatography and characterized by ^1^H-NMR, hexahydroquinazolinone (**11**) (yield 28%, Scheme 5) was obtained instead of **10**.

The ^1^H-NMR spectrum of **11** shows a *tert*-butyl signal at 0.92 ppm, *N*-Me at 3.06 ppm, and *C2* hydrogen at 4.2 ppm. The lack of an aromatic zone signals and two multiple signals between 1.41–1.74 and 2.04–2.27 ppm are of the new methylenes previously from the aromatic ring of **8**, which indicates that only partial reduction occurred.

Once the amorphous solid product **11** was characterized, it was recrystallized by diffusion overnight in *n*-hexane/CH_2_Cl_2_ (80:20) solvent mixture (Scheme 6).

Finally, after filtering, hexahydroquinazolinone (**12**) (Figure 2) was obtained whose structure was also confirmed by X-ray diffraction where a hydroperoxy group at the alpha position to the carbonyl group was noted.

As expected, the orientation of the *tert*-butyl group in a pseudo-axial position in quinazolinone (**8**) (Figure 1) causes a significant steric effect and as a consequence, which forced the hydroperoxide group to attach to the opposite side, as observed in the X-ray structure of compound **12**.

The ^1^H-NMR spectrum of the compound **12** shows the *tert*-butyl signals at 0.98 ppm, *N*-Me at 3.05 ppm, the single signal for *C2* hydrogen at 4.86 ppm, and aliphatic hydrogens between 1.55 and 2.89, respectively. In addition, from the characteristic signals of compound **12**, the ^13^C-NMR spectrum also shows the quaternary signals for *C8a* at 170.32 ppm and *C4a* at 76.32 ppm.

To confirm that the hydroperoxidized compound **12** was produced by a photooxidation reaction by exposure to ambient conditions, a kinetic study was carried out by means of ^1^H-NMR (Figure 3). Once compound **11** was purified, its ^1^H-NMR spectrum was immediately acquired (t_0_ = 0 h), and the characteristic signals of this product were noted. After 72 h (t_1_) and without opening the NMR tube, a spectrum was acquired again.

It is interesting to observe that the signals at 1.00 ppm (*10′*), 3.06 ppm (*11′*), and 4.84 ppm (*2′*) ppm at (t_1_) already increased, albeit only slightly. These signals eventually developed to be the corresponding hydrogens of **12** mentioned above. This is to show that even with the short span of light exposure before the first spectrum was obtained, the small amount of oxygen remained inside the NMR tube was enough to react with compound **11**.

Next, the NMR tube was exposed to the laboratory ambient conditions for three days with the spectrum obtained every 24 h (t_2_ = 96 h, t_3_ = 120 h, and t_4_ = 144 h). The spectrum of t_4_ shows that, by then, most of **11** was transformed into product **12**. It is important to comment that during the exposure time the volume of CDCl_3_ decreased, hence it was necessary to maintain a volume of 0.5 mL when spectra were taken.

It is also noted that, in ^1^H-NMR spectra, only signals corresponding to raw materials and products are observed, confirming the high yield and absence of competing mechanisms. This result demonstrates that the oxygen in the environment is responsible for the photooxidation reaction of compound **11** when it was recrystallized overnight.

To demonstrate that the absence of oxygen does not allow the hydroperoxidation reaction to occur, compound **11** was placed in a closed vial and after 72 h we observed only the original compound by TLC (from left, first spot with a Rf = 0.68, Figure 4). On the other hand, when **11** was placed in an open vial and covered with aluminum foil for 14 h, **12** was observed by TLC (Rf = 0.96, second spot). The third spot from the left (marked M) corresponds to a mixture of **11** and **12**, with the right spot, **12** as a reference.

To confirm the role of light in the reaction, a control TLC experiment was also performed. Compound **11** was placed in a vial under a N_2_ atmosphere and covered with aluminum foil for 24 h which resulted in only **11** was observed with Rf = 0.35 (Figure 5).

Furthermore, the reaction occurs very quickly, for even a short exposure to light and oxygen was sufficient to show traces of **12**, as can be seen in t_0_ in Figure 3. Finally, we found that with exposure to light for 6 h we were able to obtain the same result as overnight crystallization.

To confirm the importance of light exposure, it was necessary to make a detailed analysis of the kinetic study of Figure 3. Thus, Table 1 shows the ^1^H-NMR proportions of the raw material **11** and product **12**. Figure 6 illustrates the data in Table 1, where we can see that there is a sharp change in the rate of formation of **12** after 72 h where light was introduced to the vial.

Based on the experimental evidence that has been shown, a possible reaction mechanism for the formation of hydroperoxidized **12** is proposed in Scheme 7.

We argue that hydroperoxided **12** comes from the intermediate **11** through a Schenck reaction, also called an *ene* reaction, which additionally involves the singlet oxygen (^1^O_2_) as a reactant (Scheme 8) [20,21].

In a previous study [22], we showed that, in the presence of light, photoinduced elimination of the *tert*-butyl group in quinazolinones achieved a high reactivity. This gives us motive to propose (Figure 7) that **11** could also absorb energy from light to give **11*** and interact with oxygen from the environment (^3^O_2_) to give rise to the formation of singlet oxygen (^1^O_2_) through an energy transfer mechanism [23], which would result in a self-sensitizing process [24].

The general proposed pathway consists of two variants: A concerted or a stepwise mechanism [13]. The concerted pathway requires both a hydrogen atom at the allylic position and a favorable geometry in order to access the transition state kinetically [25]. Since the right-hand side ring has locked the amine hydrogen in its position, making it too rigid to be able to form a bond with the oxygen molecule, we eliminate this pathway.

For the stepwise mechanism, we explore probable intermediates following the theoretical and experimental study by Alberti and Orfanopoulos [13]. For our theoretical model, we have removed the cyclohexane ring from **11** to form **11′** (Scheme 9). The complete molecule **11** was used in a detailed study of orbital analysis, whereas model **11′** was used in order to reduce the computational cost in subsequent calculations of both locating the transition state and its confirmation by IRC.

To validate the use of **11′**, as a preliminary study we performed an orbital analysis of both **11** and **11′** at the B3LYP/6-311++G** level where we observed that **11′** shares a very similar HOMO and LUMO with **11**. This strongly suggests the use of **11′** in explore the second pathway. The product **12′** is defined in the same way.

As suggested by Alberti and Orfanopoulos [13], we consider the following intermediates: (a) biradical/dipolar, (b) perepoxide, and (c) 1,2-dioxetane (Figure 8).

Geometries of these intermediates were optimized at the same theoretical level which we found that only the Zwitterionic and the 1,2-dioxetane intermediates (labeled herein after as **I_Z_** and **I_D_** intermediates, respectively) were able to converge to stable structures whereas the perepoxide intermediate resulted in the partial dissociation of the oxygen molecule from **11′**.

In the next step, with the optimized structure of **11′** and ^1^O_2_ as reactant and **I_Z_** as product, the corresponding **TS1_Z_** were collocated by the QST2 method. This transition state is the result of a nucleophilic attack from the α-carbon of the α,β-unsaturated system to the electrophilic ^1^O_2_ (Figure 9a). The second transition state **TS1_D_**, from the same reaction of **11′** and ^1^O_2_ where **I_D_** is the product, shows the first oxygen···α-carbon bond (Figure 9b) while the nucleophilic attack from the second oxygen to the positive β-carbon to form the second bond strongly suggests a second intermediate.

Results from IRC calculations which connect a transition state to the correct reactant and product show that **TS1_Z_** leads to the intermediate **I_Z_** whereas **TS1_D_** did not lead to the reactants **11′** and ^1^O_2_ but to **I_Z_** instead. Combining the two results gives us a two-step mechanism: the first transition state leads to intermediate **I_Z_**. From there, a second transition state leads to the second intermediate **I_D_**.

In the next step, again we use the combination of QST2 and IRC methods to study the reactions between intermediates **I_Z_**, **I_D_** and the final product **12′**. For the formation of **12′** from intermediate **I_Z_** we found a six-member transition state **TS2_Z_** where a nucleophilic attack of the negatively charged oxygen to the partially positive hydrogen attached to the nitrogen atom *N*1 is observed (Figure 10). IRC calculations were used to confirm the connection between this transition state to intermediate **I_Z_** and **12′**, respectively.

The transition state of the reaction between intermediate **I_D_** and **12′**, however, is the same as **TS1_Z_** since the hydrogen atom at the *N*1 position, again due to its rigid conformation, cannot participate and, as a result, the reaction proceeds to the same **TS1_Z_** transition state.

Finally, taking into account all the hitherto theoretical and experimental evidence, we propose the following mechanism for the transformation of **11** to **12**. In the first step, the singlet oxygen molecule approaches **11** from the opposite site of the *tert*-Bu substituent, then a nucleophilic attack to the α-carbon results in an unstable dipolar/zwitterion intermediate **I_Z_** via the transition state **TS1_Z_**. The X-ray diffraction structure (Figure 2) gives evidence supporting this step. In the second step, another intermolecular nucleophilic attack from the remaining free oxygen of the peroxide to the positively-charged hydrogen (at *N*1) via the second transition state **TS2_Z_** that leads to the final product **12** (Scheme 10).

To further confirm the proposed mechanism, in which the hydrogen atom plays a crucial role, an acetylation at *N*1 by means of a Birch reduction and catalytic hydrogenation were carried out in quinazolinone **8**. Subsequently, purification of compound **11**
*N*-acylation was performed, obtaining compound **13** (Scheme 11).

Product **13** was then crystallized in the same way as for compound **11** (Scheme 12). Subsequently, the crystals was obtained and analyzed by ^1^H and ^13^C NMR, and it was confirmed that photooxidation did not occur. In the ^1^H-NMR spectrum, signals of *tert*-butyl are observed at 0.89 ppm, *N*-Me at 2.13 ppm, and a protecting group of hydrogens remains at 3.03 ppm. On the other hand, in the ^13^C-NMR spectrum, four quaternary carbons are observed: two carbonyls at 163.5 and 170.8 ppm, *C4a* at 122.3 ppm, and *C8a* 140.9 ppm.

The single crystals obtained from compound **13** were diffracted; again, the X-ray structure shows a pseudo-axial position of the *tert*-butyl group in the acquired structure (Figure 11).

We have shown that by exchanging allylic hydrogen in *N*1, the photooxidation of compound **13** does not occur under crystallization by diffusion. We suggest that the stability of hexahydroquinazolinone (**13**) increases and as a result the pair of electrons is not available to promote the nucleophilic attack of the Cα on singlet oxygen (Scheme 7).

^1^H-NMR evidence shows that the photooxidation reaction of **11** occurs in good yield. However, the corresponding yield of the precursor **11** was rather low. It has been reported [26,27,28,29,30] that endocyclic enamines can be rearomatized in the presence of catalysts, such as Pd/C, PdCl_2_, PdBr_2_, Pd(COD)_2_, Pd(OAc)_2_, or Pd(TFA)_2_. We note the enamine moiety of **11** (Scheme 1, purple box), which has been described as a possible product of the Birch reaction (step 1, Scheme 3). The same observation applies to **9a** or **9c**. These observations strongly suggest that the presence of Pd/C would accelerate the rearomatization process when catalytic hydrogenation takes place (Scheme 13). This would explain the low yields mentioned above.

## 3. Materials and Methods

Dichloromethane, ethyl acetate, and hexane were distilled before use. Toluene, acetonitrile, *tert*-butanol, isatoic anhydride, methylamine, *p*-toluene sulfonic acid, pivalaldehyde, sodium bicarbonate, sodium sulfate, sodium, palladium/carbon, 4,4-dimethylamino pyridine, and acetyl chloride were acquired from Sigma-Aldrich (St. Louis, MI, USA) and used without further purification. Reactions were monitored by thin layer chromatography (TLC) on Al plates coated with silica gel with fluorescent indicator 60 F254 (Merck-Mexico, Mexico City, Mexico). Column chromatography was performed on silica gel 60 (0.040–0.063 mm, Merck-Mexico, Mexico City, Mexico).

### 3.1. Analytical Methods

NMR spectra of products as well as the proportions of each product in the reaction mixture were recorded on Varian Gemini at 200 MHz and Varian Mercury 400 MHz (^1^H-NMR), and 50 and 100 MHz (^13^C-NMR) spectrometers, using CDCl_3_ as a solvent and tretramethylsilane (TMS) as an internal standard. A mass spectrometric analysis was performed using an Agilent 6530 quadrupole time-of-flight (QTOF) LCMS with an electrospray ionization (ESI) source (Agilent Technologies, Santa Clara, CA, USA). A mass spectrometry analysis was conducted in positive ion mode, set for a detection of mass-to-charge ratio (*m*/*z*) of 100–1000. The X-ray structures were obtained using an APEX-Bruker apparatus.

### 3.2. Theorical Study

All calculations were carried out using the Gaussian 09 suite of programs [31]. Geometry optimizations were carried out at the B3LYP/6-311++G** level of theory [32,33,34,35,36] followed by characterization by frequency calculations at the same level of the theory, in which zero-point energy corrections (ZPE) and thermal corrections at the standard state (298.15 K and 1 atm) were obtained.

Initial geometry for the model molecule **11′** was constructed from the optimized geometry of **11** by substituting the cyclohexyl moiety by suitable hydrogen atoms. Similar methodology was applied in order to obtain the geometric parameters for the product, which has taken advantage of X-ray parameters from the obtained structure. The intermediates, such as perepoxide and biradical/dipolar, were constructed from the optimized geometry of **11′** placing the O_2_ residue into the *endo* position and then optimized at the same level of theory. In order to obtain the initial coordinates for the transition state, QST2 [37] calculations were performed starting from the product and reactants’ optimized geometries. The resulting transition states were confirmed by frequency calculation. An intrinsic reaction coordinate (IRC) calculation [38] was performed at the same level of theory in order to verify that the TS structure connects with the reactant and product. Coordinates for selected structures and thermochemistry data are given in the Appendix A.

### 3.3. Chemistry

#### 3.3.1. Synthesis of 2-(*tert*-butyl)-3-methyl-2,3-dihydroquinazolin-4(1H)-one (**8**)

Quinazolinone (**8**) was prepared according to a known procedure [19] from isatoic anhydride (6 g, 30.68 mmol), methylamine in isobutanol (18.2 mL, 122.7 mmol), and ethyl acetate (50 mL). The resulting material was filtered and concentrated in a rotavapor. The crude mixture (2.06 g), *p*-toluenesulfonic acid (0.10 g, 5% *w*/*w*), pivalaldehyde (1.86 mL, 1.2 equiv.), and dichloromethane (60 mL) was refluxed for 5 h and the resulting material was filtered and evaporated under reduced pressure, purified over SiO_2_ using hexane-ethyl acetate (9:1 to 6:4). Its structure was confirmed by ^1^H and ^13^C-NMR, and compared with available information in the literature [18]. Yield: 2.71 g, 90%; white solid, mp: 145–146 °C. ^1^H NMR (200 MHz, CDCl_3_): δ 0.80 (s, 9H), 2.28 (s, 3H), 5.88 (s, 1H), 7.14–7.32 (m, 1H), 7.36 (dd, *J* = 7.6, 1.2 Hz, 1H), 7.55 (td, *J* = 7.7, 1.6 Hz, 1H), 7.67 (s, 1H), 8.02 (dd, *J* = 7.7, 1.7 Hz, 1H). ^13^C NMR (CDCl_3_, 50 MHz): δ 26.3, 38.5, 41.9, 79.9, 113.1, 116.6, 118.1, 128.1, 133.4, 146.5, 163.8. HREIMS *m*/*z* 218.1440 (calculated for C_13_H_18_N_2_O, 218.1419. Crystallographic data is deposited at Cambridge Crystallographic Data Center: CCDC no. 2006915.

#### 3.3.2. Synthesis of 2-(*tert*-butyl)-3-methyl-2,3,5,6,7,8-hexahydroquinazolin-4(1H)-one (**11**)

Ammonia (50 mL) was condensed to −78 °C before the slow addition of 1.0 g (4.6 mmol) of 2-(*tert*-butyl)-3-methyl-2,3-dihydroquinazolin-4(1H)-one (**8**), and sodium (0.64 g, 28 mmol). The resulting solution was stirred for 30 min and then treated with 1.4 mL (14 mmol) of *tert*-butanol. The reaction mixture was stirred at this temperature for 1 h. The reaction was quenched adding ammonium chloride (0.64 g, 18.5 mmol). Ammonia was evaporated at ambient temperature and reaction crude was dissolved in dichloromethane, dried with Na_2_SO_4_ and the excess solvent was evaporated under reduced pressure. A suspension of reaction crude (0.8 g), Pd/C (0.04 g, 5% *w*/*w*), and methanol (60 mL) was placed in a 100 mL flask containing a stir bar, then the system was closed with a septum. Two balloons were placed with hydrogen and the reaction was stirred overnight. The reaction mixture was filtered and concentrated in a rotavapor, then purified by column chromatography on silica, eluting with hexane-ethyl acetate (9:1 to 2:8). Yield: 130 mg, 28%; light-yellow solid; mp: 134–137 °C. ^1^H NMR (200 MHz, CDCl_3_) δ 4.18 (d, 1H, J = 4.6 Hz, 1H), 3.88 (s, 1H), 3.06 (s, 3H), 2.27–2.04 (m, 4H), 1.77–1.36 (m, 4H), 0.91 (s, 9H). ^13^C NMR of this compound was acquired for 72 h, however many signals were observed due to the high reactivity of **11**.

#### 3.3.3. Synthesis of 2-(*tert*-Butyl)-4a-hydroperoxy-3-methyl-2,4a,5,6,7,8-hexahydroquinazolin- 4(3H)-one (**12**)

Hexahydroquinazolin-4(*1H*)-one **11** (0.071 g, 0.32 mmol) was dissolved in 2 mL dichloromethane inside a suitable test tube, then 8 mL of hexane was added carefully, the system was closed with a cotton septum and left exposed to the environment overnight. Compound **12** was thus isolated as a white crystal. Yield: 81 mg (quantitative); mp: 152–154 °C; ^1^H NMR (500 MHz, CDCl_3_) δ 1.02 (s, 9H), 1.55-1.67 (m, 3H), 1.94–1.97 (m, 1H), 2.08-2.11 (m, 1H), 2.39-2.47 (m, 2H), 2.94–2.95 (m, 1H) 3.10 (s, 3H), 4.90 (s, 1H). ^13^C NMR (125 MHz, CDCl_3_) δ 170.3, 168.6, 85.3, 76.3, 40.1, 37.0, 34.2, 33.7, 28.5, 27.2, 20.5. HREIMS *m*/*z* 254.1654 (calculated for C_13_H_22_N_2_0_3_, 254.1630). Crystallographic data is deposited at Cambridge Crystallographic Data Center: CCDC no. 2006916.

#### 3.3.4. Synthesis of 1-Acetyl-2-(*tert*-butyl)-3-methyl-2,3,5,6,7,8-hexahydroquinazolin-4(1H)-one (**13**)

Hexahydroquinazolin-4(*1H*)-one (**11**) (0.8 g, 3.6 mmol), 4-dimethylaminopyridine (0.43 g, 3.6 mmol) and toluene/acetonitrile solution 9:1 (40 mL) was added to a 100 mL flask. It was placed in an ice bath and the system purged with nitrogen. Acetyl chloride (0.3 mL, 4.3 mmol) was added dropwise and the reaction was stirred overnight. The reaction crude was filtered, evaporated, and purified by column chromatography on silica, eluting with hexane-ethyl acetate (9:1 to 3:7). Yield: quantitative; white solid; mp: 146–148 °C; ^1^H NMR (200 MHz, CDCl_3_) δ 5.44 (s, 1H), 3.03 (s, 3H), 2.79–2.54 (m, 2H), 2.13 (s, 3H), 2.09–1.98 (m, 2H), 1.91–1.36 (m, 4H), 0.89 (s, 9H). ^13^C NMR (50 MHz CDCl_3_) δ 170.8, 163.5, 122.3, 38.4, 37.1, 30.5, 27.1, 27.0, 24.1, 22.6, 22.5, 21.6. HREIMS *m*/*z* 264.1843 (calculated for C_15_H_24_N_2_0_2_, 264.1838). Crystallographic data is deposited at Cambridge Crystallographic Data Center CCDC: no. 2007030.

## 4. Conclusions

We have shown that the photooxidation reaction of compound **11** occurs in the absence of sensitizers and only with exposure to ambient air. ^1^H-NMR kinetics evidence also shows that it is a process with very reasonable yield. The presence of allylic hydrogen in *N*-H bond plays an important role in the mechanism which facilitates the nucleophilic attack of α-*C* by singlet oxygen, and in the formation of the six-member cyclical transition state. The low yield of compound **11** could be explained by the aromatization reaction of enamines in the presence of Pd/C. Theoretical study supports a stepwise mechanism for the *ene* reaction between **11** and ^1^O_2_. The two-step reaction proceeds via a dipolar intermediate in which the first TS corresponds to the nucleophilic attack of the α-carbon to the ^1^O_2_ yielding the dipolar intermediate. The second step involves a nucleophilic attack from the negative oxygen to the partial positive hydrogen leading to the final product **12**. Applications of this methodology to other related 4-quinazolinone derivatives are currently being carried out in our laboratory.

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
