# Peer review of "Photooxidation of 2-(tert-Butyl)-3-Methyl-2,3,5,6,7,8-Hexahydroquinazolin-4(1H)-one, an Example of Singlet Oxygen ene Reaction†"

_molecules, 2020, doi:10.3390/molecules25215008_

Round 1

Reviewer 1 Report

In this manuscript, the authors have studied the aerobic oxidation of 2-(tert-Butyl)-3-methyl-2,3,5,6,7,8-3 hexahydroquinazolin-4(1H)-one. During the synthesis of quinazolinones, the authors observed that 2-(tert-Butyl)-3-methyl-2,3,5,6,7,8-3 hexahydroquinazolin-4(1H)-one underwent allylic hydroperoxidation under ambient conditions. They have carried out not only various experiments but also computation calculation to reveal the mechanism of allylic hydroperoxidation. On the basis of these observations, the authors proposed the plausible mechanism. Therefore, this review supports the acceptance of the present manuscript in Molecules.

1. The authors claimed photooxidation, however, they did not carry out control experiments. The experiments without light are needed to support the photooxidation.

Reviewer 2 Report

The paper by Méndez et al. describes the formation and characterization of a hydroperoxide through reaction of an enamine with ambient O2. The reaction is an interesting an important one; similar reactions may play a role in, for example, bioluminescence. However, there are some issues that need to be addressed before publication.

1.  Similar reactions have been reported in the past; though they were  not investigated computationally they should probably be cited. 

2. While the authors demonstrate the need for oxygen for the reaction to proceed; they have not clearly demonstrated the need for light. They mention that the reaction occurs in a vial covered in foil - but by the description, that seems to be a means of closure that allows oxygen to enter (i.e. not sealed) rather than a means to block light. This question is important, because free radical mechanisms occurring through triplet oxygen could also be drawn. At the least a control experiment clearly demonstrating the need for light is required. Ideally this would be supported by computational studies involving radical pathways.

 3. Though the authors report summaries of the crystal structure refinement, they should at least report full atomic coordinates for each structure.  Current standards typically require a .cif file that can be checked for consistency and errors using the IUCr checkcif website.

Reviewer 3 Report

The manuscript entitled “Photooxidation of 2-(tert-Butyl)-3-methyl-2,3,5,6,7,8- hexahydroquinazolin-4(1H)-one, an Example of Singlet Oxygen ene Reaction” describes the photooxidation of 2- (tert-butyl)-3-methyl-2,3,5,6,7,8-hexahydroquinazolin-4(1H)-one into 2-(tert-butyl)-4a-hydroperoxy-3-methyl-2,4a, 5,6,7,8-hexahydroquinazolin-4(3H)-one quantitatively without additional reactants or catalysts through the process of diffusion crystallization from n-hexane/CH2Cl2 mixture. 1H-NMR and X-ray studies has been carried out to confirm the photooxidation under investigation. A mechanism has been proposed and was confirmed by control experiments, as well as theoretical calculations.

All in all, the work is well executed and discussed and the manuscript is well written. A couple of points that needs to be addressed are as follows:

  1. Starting from compound 8, compound 11 was obtained in 28% yield, while compound 13 was obtained in quantitative yield, is there any justification for that? is because of an incomplete reaction or the formation of a by-product?
  2. High quality images for Schemes 5 and 6 are required.
  3. For Scheme 2, is all the products displayed in that scheme are formed together (obtained as a mixture)? It is not clear as the "+" signs are between cis-4 and trans-6 and between cis-4 and cis-5, only.

I would recommend publication after minor revision.

Round 2

Reviewer 2 Report

The authors have addressed the main problems indicated in my earlier review - however their description of the control experiment is still deficient. To determine  if the reaction is truly a photooxidation, (rather than a dark reaction) the control should include the sample and oxygen but in the absence of light. The control the authors describe involves the sample under an N2 atmosphere in the absence of light. Without oxygen present, it is clear that any kind of oxygenation reaction, photochemical or otherwise, would be impossible and thus the control they describe is not sufficient.

I should add that I consider the reaction the authors describe to be interesting enough for publication whether the reaction is photochemical or not. But it is important that the designation as a photochemical reaction is adequately justified.

Author Response

Referee suggestion was attended (see p.6, Lines 164-175) and also Table 1 and Figure 6 were added to give greater clarity for the experiments that support the photooxidation reaction.

Now we have added the paragraf that states:

“To confirm the importance of light exposure, it was necessary to make a detailed analysis of the kinetic study of Figure 3. Thus, Table 1 shows the 1H-NMR proportions of the raw material 11 and product 12. Figure 6 illustrates the data in Table 1, where we can see that there is a sharp change in the rate of formation of 12 after 72 h where light was introduced to the vial.

Round 3

Reviewer 2 Report

The authors have now corrected the paper to my satisfaction. The new additions clearly illustrate the significance of light exposure.

The paper can be published as is.